# Improving Mammography Malignancy Segmentation with a Semi-Supervised Training Process

**Mickael Tardy**[1,2]                    MICKAEL.TARDY@EC-NANTES.FR
[1] *Ecole Centrale de Nantes, LS2N, UMR CNRS 6004, Nantes, France*
[2] *Hera-MI, SAS, Nantes, France*

**Diana Mateus**[1]                        DIANA.MATEUS@EC-NANTES.FR

## Abstract

We work on the breast imaging malignancy segmentation task while focusing on the training process instead of network complexity. We designed a training process based on a modified U-Net, increasing the overall segmentation performances by using both, benign and malignant data for training. Our approach makes use of only a small amount of annotated data and relies on transfer learning from a self-supervised reconstruction task, and favors explainability.

**Keywords:** Mammography, Segmentation, Malignancy Detection, Explainability

## 1. Introduction

Breast cancer is one of the most frequent cancers, and thus, an important public health issue. Mammography is the most common screening exam in early breast cancer detection. While a big amount of imaging is generated, its use for learning-based CAD solutions is not obvious due to the lack of detailed annotations of the malignant regions. In this work, we propose a method to train a segmentation network having few annotated images.

Typically, to perform image segmentation with a supervised deep learning approach, the network needs an explicit segmentation mask as a ground truth. For medical imaging, this means that each image shall have delimited regions of interest provided by an expert. In clinical practice, such ground truth is rarely available since its collection is a tedious and time-consuming process. Besides, only malignant images can be labeled pixel-wise, which are only a portion of the available data. However, even with 1/8 of women population being affected by breast cancer (Siegel et al., 2019), a great part of mammography imaging is benign and can not be used for supervised segmentation training.

We address the problem with a two-step training process. In first, we follow-up on the work of (Zhou et al., 2019), which proposes weights initialization by pre-training the neural network for reconstruction in a self-supervised manner, before training it for the same or different application. Such an approach enables taking advantage of large amounts of benign images to initialize the model with prior knowledge, improving the overall performance. As in (Zhou et al., 2019), we train a reconstruction network in a self-supervised manner but adapt it to full-images instead of patches (see Section 2). In our second step, we wrapped the backbone U-Net with a subtraction layer on top of the U-Net (see Fig. 1), taking the output from the input, so the network is still trained for reconstruction, rather than being trained to yield the probability of pixel belonging to a malignant region.

## 2. Methods

Our training process (see Fig. 1) is composed of two steps: i) image-wise self-supervised training on full-sized benign mammograms for the reconstruction task followed by ii) image-wise discriminative reconstruction training on malignant mammograms.

**U-Net for full-size mammograms** We trained the network on the full images (mammograms cropped and rescaled to **1536x1536**), instead of patches. The cropping allows us to keep the pixel spacing low and the smaller findings visible. We modified traditional U-Net (Ronneberger et al., 2015) as follows: i) we used separable convolutions (Chen et al., 2018; Qi et al., 2019), ii) we adjusted the skip connections, removing the top long ones, and introducing short ones in each block (Drozdzal et al., 2016; Szegedy et al., 2016), iii) we used instance normalization and leaky ReLU activations (Isensee et al., 2019).

**Self-supervised training** We adopted the self-supervised training approach (Zhou et al., 2019), with few modifications. Unlike Zhou et al. who focused on patch-wise trining of CT images, we worked with image-wise training on mammograms, so we introduced operations that better fit the task, namely: i) instead of non-linear intensity transformation we used gamma-based one, ii) we excluded local-shuffling as too harmful for full mammograms, and iii) we extended in-painting with a range of non-rectangle shapes.

At this stage, we use only benign mammograms, filtered by the patient-wise labels: that is, only confirmed benign patient-files were included. We filtered the dataset according to the patient-wise labels of ACR cancer probability. We kept only the cases classified as ACR1 and ACR2 and remove all the others (ACR3-6) as uncertain or malignant.

**Image-wise segmentation** Once the U-Net network was initialized with the reconstruction task on benign mammograms, we trained it on the malignant images (ACR4-6) in a supervised manner with pixel-wise ground truth. However, instead of changing the top activation of the U-Net, we added a layer on top of the U-Net (see Fig. 1), subtracting the output from the input. Thus, we continue to train the backbone network for a reconstruction task, while the whole model was designed to yield malignancy segmentation regions (see Fig. 2).

## 3. Results

We show the performances on the malignant images of INBreast dataset (Moreira et al., 2012) in Table 1 (80% train, 20% test). We note, that unlike common U-Net implementations (Sun et al., 2020), we combine both, masses and calcification masks.

Table 1: Segmentation performances

| Method | $DICE_{Max}$ | $DICE_{Avg}$ (on 10 eps) |
|---|---|---|
| Fully Supervised segmentation (*from scratch*) (baseline) | 0.58 | 0.44 |
| Fully Supervised seg. (*pre-trained*) (Zhou et al., 2019) | 0.58 | 0.57 |
| **Discriminative segmentation (*from scratch*) (ours)** | **0.61** | 0.52 |
| **Discriminative segmentation (*with pre-training*) (ours)** | **0.61** | **0.59** |

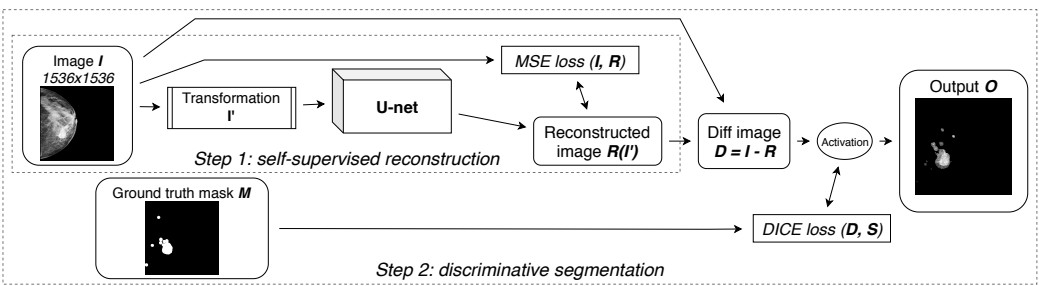

Figure 1: Two-step training process design

## 4. Discussion

Our network can be trained on both, benign and malignant images. Such an approach yields better segmentation performances (i.e. higher $DICE_{Max}$) and acts as a natural regularizer (i.e. higher $DICE_{Avg}$) since the transition to malignant images is closer to fine-tuning than to transfer learning. Our results show that such duality is beneficial for the segmentation results (see Tab. 1). Working on high-resolution mammograms, our discriminative network is sensitive to both, masses and calcifications (see Fig. 2).

Our approach contributes to the explainability of the output: our network is designed to discriminate malignant regions on the full-sized mammograms instead of yielding a probability of pixel to belong to a malignant region. Our network is trained with full mammograms, so, the spatial information is kept by design. The proposed process is scalable, meaning it can take advantage of both benign and malignant imaging. Furthermore, such a process more naturally integrates with the federated or distributed learning scheme thanks to its efficient data use, thus its ability to switch easier between vendors.

The obtained results may be helpful to the radiologists when performing diagnosis. It also can help radiographers making additional examinations (such as spot localizations or magnification), before interpretation by a radiologist.

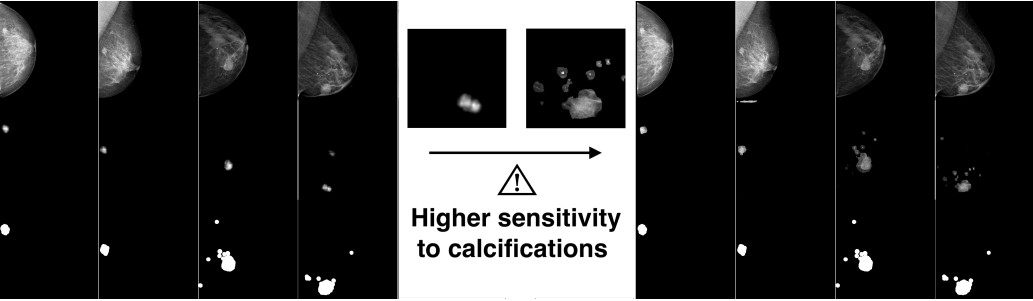

Figure 2: Segmentation results. **Left**: U-Net (pre-trained), **Right**: Discriminative network (pre-trained). **1st row**: Input $I$, **2nd row**: Output $O$, **3rd row**: GT masks $M$

## Acknowledgments

Research funding is provided by Hera-MI, SAS and Association Nationale de la Recherche et de la Technologie, CIFRE grant no. 2018/0308

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
