# OpenReview forum: "Improving Mammography Malignancy Segmentation by Designing the Training Process"
_MIDL.io/2020/Conference — MIDL 2020_

### Official Review · AnonReviewer4 · 2020-03-13
**Strange method with no apparent use.**

**Rating:** 1
**Confidence:** 5

**Review:**

The authors claim 1/8 screening mammograms contain a malignant lesion. This is obviously false. It is also unclear why segmentation would be an important task in screening. The goal is to select images with suspicious findings, ensuring not too many false negatives yet achieve a good positive predictive value. Many of the lesions one wants to detect are calcified and cannot be trivially segmented. Therefore, it is surprising that as a preprocessing step, the images are rescaled to a resolution of 1536 x 1536. For these calcified lesions, this can remove a lot of structure (and ignoring that these images have all different aspect ratios).

While the method might be interesting to show, there are too many flaws (explanation of the method, details of the dataset, details on training and so on) and too little context that it would warrant acceptance.

---

### Official Review · AnonReviewer2 · 2020-03-14
**Too many details missing**

**Rating:** 2
**Confidence:** 5

**Review:**

- The general idea of this paper is to design a training process to improve segmentation.

- The authors propose to incorporate the unlabelled data with self-supervised learning into the training process.

- I think the paper is well-motivated, but too many details are missing, making it not possible to understand.

---

### Official Review · AnonReviewer1 · 2020-03-16
**Self-supervision to initialize segmentation model for mammography**

**Rating:** 3
**Confidence:** 3

**Review:**

The paper proposed self-learning to pre-train a U-Net model using benign cases, that would not be otherwise used because not containing annotations, to perform a reconstruction task. This U-Net is later used to segment lesions using manual annotations. A public dataset is used for evaluation. The use of self-supervision is interesting here, and shows some improvement whe compared to using full supervision. In the evaluation, masses and calcifications are merged, which may have an impact in the clinical applicability of this system. No comparison with existing models on the same public dataset are reported.

---

### Meta-Review · Area_Chair1 · 2020-03-27
**MetaReview of Paper137 by AreaChair1**

**Rating:** 3

**Metareview:**

The paper tries to incorporate the unlabelled data with self-supervised learning into the training process.
The major issue is lacking of the details. But since it is a short paper, so I suggest it for weak acceptance due to its novelty in the training process.
The citations need to be properly updated to use the journal publication ones instead of archives. Please carefully go through the citations.

**Paper Type:**

validation/application paper

---

### Decision · Program_Chairs · 2020-04-11

Accept